# E-Bayesian Estimation for the Weibull Distribution under Adaptive Type-I Progressive Hybrid Censored Competing Risks Data

**DOI:** 10.3390/e22080903

**Published:** 2020-08-17

**Authors:** Hassan Okasha, Abdelfattah Mustafa

**Affiliations:** 1Department of Statistics, Faculty of Science, King Abdulaziz University, Jeddah 21589, Saudi Arabia; 2Department of Mathematics, Faculty of Science, Al-Azhar University, Cairo 11884, Egypt; 3Department of Mathematics, Faculty of Science, Islamic University of Madinah, Madinah 42351, Saudi Arabia; amelsayed@mans.edu.eg; 4Department of Mathematics, Faculty of Science, Mansoura University, Mansoura 35516, Egypt

**Keywords:** adaptive type-I progressive hybrid censored, competing risks, cumulative exposure model, Bayesian estimation, E-Bayesian estimation, E-mean-square error

## Abstract

This article focuses on using E-Bayesian estimation for the Weibull distribution based on adaptive type-I progressive hybrid censored competing risks (AT-I PHCS). The case of Weibull distribution for the underlying lifetimes is considered assuming a cumulative exposure model. The E-Bayesian estimation is discussed by considering three different prior distributions for the hyper-parameters. The E-Bayesian estimators as well as the corresponding E-mean square errors are obtained by using squared and LINEX loss functions. Some properties of the E-Bayesian estimators are also derived. A simulation study to compare the various estimators and real data application is applied to show the applicability of the different estimators are proposed.

## 1. Introduction

In life-testing and reliability studies, both type I and type II censoring schemes are widely used. These two types of censoring schemes are described as follows: Consider *n* identical components are placed in the test, in type I censoring, the experiment continues up to a predetermined time τ. However, in the type II censoring scheme, the experiment is terminated when a predetermined number of failures m<n occurs. For a mixture of type I and type II censoring schemes, which is called the type I hybrid censoring scheme, is introduced by Epstein [1], and the life test experiment is terminated at a random time $\tau^{*}=\min\left\{x_{m:m:n},\tau \right\}$. Childs et al. [2] proposed a new hybrid censoring scheme called a type-II hybrid censoring scheme in which the experiment would terminate at the random time $\tau^{*}=\max\left\{x_{m:m:n},\tau \right\}$. These schemes do not allow for removing the components from the experiment at any time other than the terminal point. A more general censoring scheme called progressive type II censoring is used to deal with this problem.

In the progressive type II censoring, *n* components are placed in a lifetime testing experiment, and *m* is a fixed number of components to be failed. At the time of the first failure $x_{1:m:n}$, $R_{1}$ components are randomly removed from the remaining n−1 components. Similarly, at the time of the second failure $x_{2:m:n}$, $R_{2}$ components of the remaining $n-2-R_{1}$ components are randomly removed, and so on. At the time of the *m*-th failure $x_{m:m:n}$, all the remaining $n-m-\sum_{i=1}^{m-1}R_{i}$ components are removed. The $R_{i},i=1,\cdots ,m$ are fixed and predetermined prior to the study.

The type I progressive hybrid censoring scheme is considered by Kundu and Joarder [3], which is a mixture of type II progressive and hybrid type I censoring schemes—in which *n* identical components are put under testing with $R_{i},i=1,\cdots ,m$, predetermined progressive censoring scheme and the experiment is terminated at random time $\tau^{*}=\min\left\{x_{m:m:n},\tau \right\}$. The experiment stops at the time $x_{m:m:n}$, if the *m*-th failure occurs before the time τ. However, if the *m*-th failure does not occur before time τ and only *J* failures occur before the time τ; then, at the time τ, the experiment terminates and all the remaining components are removed. In addition, a type II progressive hybrid censoring scheme is discussed by Childs et al. [4], where the experiment is terminated at time $\tau^{*}=\max\left\{x_{m:m:n},\tau \right\}$.

A new censoring scheme, called an adaptive type II progressive hybrid censoring, is introduced by Ng et al. [5], where the number of failures *m* and the corresponding progressively scheme is given, but no components will be removed when the experimental time passes time τ; see, for example, Balakrishnan and Kundu [6]. Another adaptive progressive hybrid censoring scheme, called the adaptive type I progressive hybrid censoring scheme (AT-I PHCS), which assures the termination of the lifetime testing experiment at a fixed time τ, and results in a higher efficiency in estimations, is proposed by Lin and Huang [7]. The AT-I PHCS can be described as: suppose *n* identical components are placed under testing with prefixed $R_{i},i=1,\cdots ,m$, 1≤m≤n; the experiment is terminated at a prefixed time τ, where τ∈(0,∞). At the time $x_{1:m:n}$, $R_{1}$ of the remaining components are randomly removed, at the time $x_{2:m:n}$, $R_{2}$ of the remaining components are randomly removed, and so on. Let the number of failures that occur before time τ be *J*. If the *m*-th failure $x_{m:m:n}$ occurs before time τ, the process will not stop, but continue to observe failures without any further withdrawals until reach time τ. Then, all remaining components $R_{J}^{*}=n-J-\sum_{i=1}^{J}R_{i}$ are removed at time τ, and the experiment is terminated. The progressive censoring scheme in this case will become $R_{1},R_{2},\cdots ,R_{m},R_{m+1},\cdots ,R_{J}$, where $R_{m}=R_{m+1}=\cdots =R_{J}=0$. Otherwise, when $x_{m:m:n}>\tau$, the process will have a progressive censoring scheme such as $R_{1},R_{2},\cdots ,R_{J}$. AT-I PHCS is important when the time is the main goal in the experiment, and it is requisite to terminate the experiment at a predetermined time in any case of number of failures.

The point and interval estimation for the exponential distribution are studied by Lin and Huang [7] and investigated Bayesian sampling plans under different progressive censoring schemes. The maximum likelihood and Bayesian estimation for a two-parameter Weibull distribution based on AT-I PHCS are discussed by Lin et al. [8]. They derived the Bayes estimates of the unknown parameters by using the approximated form of Lindley [9] and Tierney and Kadane [10].

On the other hand, the loss function is important in Bayesian methods. In the Bayesian inference, the most commonly used loss function is the squared error loss. It is well known that the use of symmetric loss functions may be inappropriate in many circumstances, particularly when positive and negative errors have different consequences. One of the most commonly used asymmetric loss functions is the LINEX (linear exponential) loss function. It was introduced by Varian [11] became popular due to Zellner [12].

The failure of components in reliability analysis at the same time may be attributable to more than one reason. These reasons are competing for the experimental component for the failure. This is known as the competing risks model. The data in the competing risks analysis are comprised of a failure time and the associated reason for failure. The reasons for failure can be assumed to be independent or dependent.

The latent failure times model in this paper is assumed as suggested by Cox [13]. In addition, the failure times are independently distributed. The failure is due to more than one reason, see Crowder [14]. The competing risks data discussed here under AT-I PHCS and Weibull distributions with common shape parameters are assumed to the failure times. The competing risks data under AT-I PHCS with the assumption of exponential distribution are analyzed by Ashour and Nassar [15]. The exponential distribution has a constant failure rate, so it has serious limitations in modeling lifetime data. The inference for Weibull distribution under adaptive type-I progressive hybrid censored competing risks data are investigated by Ashour and Nassar [16].

The main aim of this paper is studying the competing risk model under AT-I PHCS. The lifetimes under the competing risks have independent Weibull distributions with common shape parameters. This paper can be organized as follows:

The model description and the notation are introduced in Section 1. The maximum likelihood estimation of the unknown parameters is established in Section 3. Bayesian estimation of the parameters under squared error (SE) and LINEX loss functions are discussed in Section 4. The expected Bayesian estimation under squared error and LINEX loss functions are derived in Section 5. Some properties of the E-Bayesian estimators are also derived in Section 6. Finally, two examples of the real data set and numerical simulation results are presented in Section 7.

## 2. Model Description

Suppose *n* identical components are put into a lifetime test with prefixed progressive censoring scheme $R_{i},i=1,\cdots ,m,$
1≤m≤n and the experiment is terminated at time τ, where τ∈(0,∞). The lifetime for the components are assumed to be the Weibull distribution. Under the adaptive type-I progressive censoring scheme and in the presence of competing risks data, we have the following observation:(1)$$\left(X_{1:m:n},c_{1},R_{1}\right),\cdots ,\left(X_{m-1:m:n},c_{m-1},R_{m-1}\right),\left(X_{m:m:n},c_{m},0\right),\cdots ,\left(X_{J:m:n},c_{J},0\right),$$
where the indicator $c_{i}$ is denoting the reason of failure, and *J* is the number of failures before time τ.

Consider that $R_{J}^{*}$ is the number of remaining components left at the time point τ with $R_{m}=R_{m+1}=\cdots =R_{J}=0$. Let $c_{i}\in \left(1,2\right)$, here, $c_{i}=k,k=1,2$ means that the component *i* has failed due to reason *k*. Furthermore, we define
$$I_{1}\left(c_{i}=1\right)=\left\{\begin{array}{cc} 1, & c_{i}=1\\ 0, & \mathrm{otherwise} \end{array}\right.\;\;\mathrm{and}\;\;I_{2}\left(c_{i}=2\right)=\left\{\begin{array}{cc} 1, & c_{i}=2\\ 0, & \mathrm{otherwise} \end{array}\right.$$

Thus, the random variables $D_{k}=\sum_{i=1}^{J}I_{1}\left(c_{i}=k\right)$ describe the number of failures due to the reason k,k=1,2 for failure.

The latent failure times $X_{1i}$ and $X_{2i}$ are assumed to be independent. $X_{ki},i=1,\cdots ,n$, have Weibull distributions with parameters β and $\lambda_{k},\left(k=1,2\right)$ (same shape and different scale parameter). The corresponding survival function ${\bar{F}}_{k}$ and the hazard rate function $h_{k}$ are given, respectively, by
(2)$${\bar{F}}_{k}\left(x;\lambda_{k},\beta \right)=e^{-\lambda_{k}x^{\beta }},\;\;\;\;\;x>0,\lambda_{k},\beta >0,$$
and
(3)$$h_{k}\left(x;\lambda_{k},\beta \right)=\lambda_{k}\beta x^{\beta -1},\;\;\;\;\;x>0,\lambda_{k},\beta >0.$$

## 3. Maximum Likelihood Estimation

Based on adaptive type-I progressive hybrid censoring with a sample of size *m* obtained from a life test experiment of *n* items from the Weibull distribution, the likelihood function of the observed data $\left(x_{1},c_{1}\right),\cdots ,\left(\tau ,R_{J}^{*}\right)$, for a given scheme $R_{1},R_{2},\cdots ,R_{m-1},0\cdots ,0,R_{J}^{*}$, can be written as
(4)$$L=c\prod_{i=1}^{J}\left\{{\left[f_{1}\left(x_{i}\right){\bar{F}}_{2}\left(x_{i}\right)\right]}^{I(c_{i}=1)}{\left[f_{2}\left(x_{i}\right){\bar{F}}_{1}\left(x_{i}\right)\right]}^{I(c_{i}=2)}{\left[{\bar{F}}_{1}\left(x_{i}\right){\bar{F}}_{2}\left(x_{i}\right)\right]}^{R_{i}}\right\}{\left[{\bar{F}}_{1}\left(\tau \right){\bar{F}}_{2}\left(\tau \right)\right]}^{R_{J}^{*}},$$
where $x_{i}=x_{i:m:n}$ and *c* is a constant that doesn’t depend on the parameters.

The likelihood function can be represented by applying the identity $f_{k}=h_{k}{\bar{F}}_{k}$ as follows:(5)$$L=c\prod_{i=1}^{J}\left\{{\left[h_{1}\left(x_{i}\right)\right]}^{I(c_{i}=1)}{\left[h_{2}\left(x_{i}\right)\right]}^{I(c_{i}=2)}{\left[{\bar{F}}_{1}\left(x_{i}\right){\bar{F}}_{2}\left(x_{i}\right)\right]}^{R_{i}+1}\right\}{\left[{\bar{F}}_{1}\left(\tau \right){\bar{F}}_{2}\left(\tau \right)\right]}^{R_{J}^{*}},$$
where $J=D_{1}+D_{2}$ and J>0.

From Equations (Equation 2), (Equation 3) and (Equation 5), the likelihood function ignoring the normalized constant can be written as follows:(6)$$\begin{array}{ccc} L(\lambda_{1},\lambda_{2}|\underline{x}) & = & \lambda_{1}^{D_{1}}\lambda_{2}^{D_{2}}\varphi \left(\beta ;\underline{x}\right)e^{-\left(\lambda_{1}+\lambda_{2}\right)T}, \end{array}$$
where $\underline{x}=(x_{1},x_{2},\cdots ,x_{r})$, $\varphi \left(\beta ;\underline{x}\right)=\beta^{J}\prod_{i=1}^{J}x_{i}^{\beta -1}$, $T=\sum_{i=1}^{J}\left(1+R_{i}\right)x_{i}^{\beta }+R_{J}^{*}\tau^{\beta }$, $R_{J}^{*}=n-J-\sum_{i=1}^{J}R_{i}$, and $x_{i}=x_{i:m:n}$ for simplicity of notation.

Assume that the parameter β is known and takes the natural logarithm of (Equation 6); then, differentiating with respect to $\lambda_{1}$ and $\lambda_{2}$, we can obtain the MLEs of $\lambda_{1}$ and $\lambda_{2}$ as
(7)$${\hat{\lambda }}_{1}=\frac{D_{1}}{T}\;\;\;\mathrm{and}\;\;\;{\hat{\lambda }}_{2}=\frac{D_{2}}{T}.$$

Before progressing further, we assume the case of $1\leq D_{i}\leq m$, i=1,2 in all of the next sections. Three ways to get an estimation performance assessment are the expectation, variance, and mean-square error (MSE). It can be shown that *T* has a Gamma ($D_{i},\lambda_{i}$, i=1,2) distribution. Therefore, the expectations and mean square errors for all estimators mentioned above are obtained respectively as follows: For MLE, ${\hat{\lambda }}_{{}_{MLE}}$, of λ,
$$E\left({\hat{\lambda_{i}}}^{MLE}\right)=\int_{0}^{\infty }\frac{D_{i}}{t}\frac{\lambda_{i}^{D_{i}}}{\Gamma (D_{i})}t^{D_{i}-1}e^{-\lambda_{i}t}dt=\frac{D_{i}}{D_{i}-1}\lambda_{i},\;i=1,2,$$
and
(8)$$\begin{array}{cc} MSE\left({\hat{\lambda_{i}}}^{MLE}\right) & =E{\left(\lambda_{i}-{\hat{\lambda_{i}}}^{MLE}\right)}^{2}=\lambda_{i}^{2}-2\lambda_{i}E\left({\hat{\lambda_{i}}}^{MLE}\right)+E{\left({\hat{\lambda_{i}}}^{MLE}\right)}^{2}\\ & =\frac{D_{i}+2}{\left(D_{i}-1\right)\left(D_{i}-2\right)}\lambda_{i}^{2},\;i=1,2. \end{array}$$

## 4. Bayesian Estimation

In this section, we obtain the Bayes estimators of the parameters $\lambda_{1}$ and $\lambda_{2}$ based on SE and LINEX loss functions. For developing the Bayesian estimation, we assume that the parameters $\lambda_{1}$ and $\lambda_{2}$ are independently distributed according to gamma distribution. Let $\lambda_{j},j=1,2$ have a gamma prior with scale parameters $c_{j}$ and shape parameters $d_{j}$. The joint prior density of $\lambda_{1}$ and $\lambda_{2}$ can be written as follows:(9)$$g\left(\lambda_{1},\lambda_{2}\right)\propto \lambda_{1}^{d_{1}-1}\lambda_{2}^{d_{2}-1}e^{-c_{1}\lambda_{1}-c_{2}\lambda_{2}},\;c_{j},d_{j}>0,j=1,2.$$

The joint prior (Equation 9), can be derived as a special case from the dependent prior proposed by Kundu and Pradhan [17]. Based on the likelihood function (Equation 6) and the joint prior density (Equation 9), the joint posterior density of $\lambda_{1}$ and $\lambda_{2}$ given β, and the data can be written as
(10)$$q\left(\lambda_{1},\lambda_{2}|\underline{x}\right)\propto \lambda_{1}^{D_{1}+d_{1}-1}\lambda_{2}^{D_{2}+d_{2}-1}e^{-\lambda_{1}\left(T+c_{1}\right)-\lambda_{2}\left(T+c_{2}\right)}.$$

From (Equation 10), we observe that the posterior density functions of $\lambda_{1}$ and $\lambda_{2}$ are $gamma(D_{1}+d_{1},T+c_{1})$ and $gamma(D_{2}+d_{2},T+c_{2})$, respectively. Based on the SE loss function, the Bayes estimators of $\lambda_{1}$ and $\lambda_{2}$ can be obtained as the posterior means with the following forms:(11)$${\hat{\lambda }}_{j}^{BS}=\frac{D_{j}+d_{j}}{T+c_{j}},\;j=1,2.$$

Considering $d_{1}=d_{2}=c_{1}=c_{2}=0$, the Bayes estimators in (Equation 11) coincide with the MLEs in (Equation 7).

Now, we obtain the Bayes estimators based on LINEX loss function proposed by Varian [11]. The LINEX loss function with parameter ω is given by
(12)$$\psi \left(\mu^{BL},\mu \right)=e^{\omega (\mu^{BL}-\mu )}-\omega \left(\mu^{BL}-\mu \right)-1,$$
where ω≠0 is a constant, μ is the parameter to be estimated, and $\mu^{BL}$ is the Bayes estimator under the LINEX loss function. From (Equation 12), the Bayes estimator $\mu^{BL}$ is
(13)$$\mu^{BL}=-\frac{1}{\omega }\ln E\left(e^{-\omega \mu }\right),\;\;\;\omega \neq 0.$$

From (Equation 10) and (Equation 13), the Bayes estimators of the parameters $\lambda_{1}$ and $\lambda_{2}$ can be obtained as
(14)$${\hat{\lambda }}_{j}^{BL}=\frac{D_{j}+d_{j}}{\omega }\ln\left(1+\frac{\omega }{T+c_{j}}\right),\;j=1,2.$$

Under the SE loss function, the MSE of the parameters, $\lambda_{j}$, j=1,2, can be obtained as
(15)$$\begin{array}{cc} MSE\left({\hat{\lambda }}_{j}^{BS}\left(c_{j},d_{j}\right)|T\right) & =E\left({\left(\lambda_{j}-{\hat{\lambda }}_{j}^{BS}\left(c_{j},d_{j}\right)\right)}^{2}|T\right)\\ \; & =E\left(\lambda_{j}^{2}|T\right)-2{\hat{\lambda }}_{j}^{BS}\left(c_{j},d_{j}\right)E\left(\lambda_{j}|T\right)+{\left({\hat{\lambda }}_{j}^{BS}\left(c_{j},d_{j}\right)\right)}^{2}\\ \; & =\frac{\left(D_{j}+d_{j}+1\right)\left(D_{j}+d_{j}\right)}{{\left(T+c_{j}\right)}^{2}}-2\left(\frac{D_{j}+d_{j}}{T+c_{j}}\right)\left(\frac{D_{j}+d_{j}}{T+c_{j}}\right)+{\left(\frac{D_{j}+d_{j}}{T+c_{j}}\right)}^{2}\\ \; & =\frac{D_{j}+d_{j}}{{\left(T+c_{j}\right)}^{2}},\;j=1,2. \end{array}$$

In addition, when $-1<\frac{w}{T+c_{j}}<1$, the Bayesian estimator, ${\hat{\lambda }}_{j}^{BL}\left(c_{j},d_{j}\right)$, j=1,2, has the following properties:(16)$$\begin{array}{cc} MSE\left({\hat{\lambda }}_{j}^{BL}\left(c_{j},d_{j}\right)|T\right) & =E\left({\left(\lambda_{j}-{\hat{\lambda }}_{j}^{BL}\left(c_{j},d_{j}\right)\right)}^{2}|T\right)\\ & =E\left(\lambda_{j}^{2}|T\right)-2{\hat{\lambda }}_{j}^{BL}\left(c_{j},d_{j}\right)E\left(\lambda_{j}|T\right)+{\left({\hat{\lambda }}_{j}^{BL}\left(c_{j},d_{j}\right)\right)}^{2}\\ & =\frac{\left(D_{j}+d_{j}+1\right)\left(D_{j}+d_{j}\right)}{{\left(T+c_{j}\right)}^{2}}-2\frac{{\left(D_{j}+d_{j}\right)}^{2}}{w(T+c_{j})}\ln\left(1+\frac{w}{T+c_{j}}\right)\\ & \;\;\;\;+{\left(\frac{D_{j}+d_{j}}{w}\ln\left(1+\frac{w}{T+c_{j}}\right)\right)}^{2}\\ & =\frac{D_{j}+d_{j}}{{\left(T+c_{j}\right)}^{2}}+{\left(\frac{D_{j}+d_{j}}{T+c_{j}}\right)}^{2}{\left\{\sum_{i=2}^{\infty }\frac{w^{\left(i-1\right)}{\left(-1\right)}^{\left(i-1\right)}}{i{\left(T+c_{j}\right)}^{\left(i-1\right)}}\right\}}^{2},\;j=1,2. \end{array}$$

## 5. E-Bayesian Estimation

E-Bayesian (Expected Bayesian) estimation was first introduced in literature by Han [18]. He obtained the estimate of the scale parameter of the Weibull distribution based on SE loss function and also derived the properties of the E-Bayesian estimation. E-Bayesian based on three different prior distributions of hyper parameter are used in this section to investigate the influence of different prior distributions on the E-Bayesian of $\lambda_{j},j=1,2$. For more relevant research about the E-Bayesian estimation, see Han [19], Jaheen and Okasha [20], Azimi et al. [21], Okasha [22], Okasha and Wang [23], and Abdallah and Jumping [24].

Han [18] stated that the prior distribution of $d_{j}$ and $c_{j},j=1,2$ should be determined to ensure that the prior distribution $g(\lambda_{j}),j=1,2$ is a decreasing function in $\lambda_{j},j=1,2$. To be sure from this condition, we find the first derivative of $g(\lambda_{j}),j=1,2$ with respect to $\lambda_{j},j=1,2$ as
$$\frac{\partial g(\lambda_{j})}{\partial \lambda_{j}}\propto \lambda_{j}^{d_{j}-2}e^{-c_{j}\lambda_{j}}\left[d_{j}-1-c_{j}\lambda_{j}\right],\;j=1,2.$$

Thus, for $0<d_{j}<1$ and $c_{j}>0,j=1,2$, the function $\frac{\partial g(\lambda_{j})}{\partial \lambda_{j}}<0,j=1,2,$ and therefore $g(\lambda_{j}),j=1,2$ is a decreasing function of $\lambda_{j},j=1,2$. Suppose that $d_{j}$ and $c_{j},j=1,2$ are independent with bivariate PDF given by
$$\pi \left(d_{j},c_{j}\right)=\pi_{j}\left(d_{j}\right)\pi_{j}\left(c_{j}\right),\;j=1,2.$$

Then, the E-Bayesian estimate of the parameter $\lambda_{j},j=1,2$, (expectation of the Bayesian estimate of $\lambda_{j},j=1,2$), according to Han [18] can be obtained as follows:(17)$${\hat{\lambda_{j}}}^{EB}=\int \int_{D}{\hat{\lambda_{j}}}^{B}\left(d_{j},c_{j}\right)\pi \left(d_{j},c_{j}\right)dd_{j}dc_{j},\;j=1,2,$$
where ${\hat{\lambda_{j}}}^{B}\left(d_{j},c_{j}\right)$ is the Bayesian estimator of $\lambda_{j}$ given by Equations (Equation 11) and (Equation 14), ${\hat{\lambda_{j}}}^{EB}\left(d_{j},c_{j}\right)$ is the expected Bayes estimate of $\lambda_{j}$, j=1,2 under any loss function. The value range of hyper parameter $d_{j}$ and $c_{j},j=1,2$ satisfy $D=\left\{\left(a,b\right)|0<d_{j}<1,0<c_{j}<s,j=1,2\right\}$. Suppose the prior distribution of $d_{j}$ and $c_{j},j=1,2$ are beta distribution and uniform distribution in (0,s), respectively. For more details about E-Bayesian estimation, see Han [18], Okasha [22] and Okasha, and Wang [23].

The expected mean square error (E-MSE) of the parameter $\lambda_{j},j=1,2$ according to Han [19] can be obtained as follows:(18)$$E-\mathrm{MSE}\left(\hat{\lambda_{j}}\right)=\int \int_{D}MSE\left(\hat{\lambda_{j}}\right)\left(d_{j},c_{j}\right)\pi \left(d_{j},c_{j}\right)dd_{j}dc_{j},\;j=1,2,$$
where $MSE\left(\hat{\lambda_{j}}\right)\left(d_{j},c_{j}\right)$ is the MSE of $\lambda_{j}$, j=1,2 under any loss function.

### 5.1. E-Bayesian Estimation under SE Loss Function

Based on three different prior distributions of the hyper-parameters $d_{j}$ and $c_{j}$, the E-Bayesian estimates of the parameter $\lambda_{j},j=1,2$ can be obtained. Accordingly, these prior distributions are selected to show the effect of the different prior distributions on the E-Bayesian estimation of the parameter $\lambda_{j},j=1,2$. The selected priors distributions are given by
(19)$$\left.\begin{array}{cc} \pi_{1}\left(d_{j},c_{j}\right)=\frac{1}{sB(a,b)}d_{j}^{a-1}{\left(1-d_{j}\right)}^{b-1} & 0<d_{j}<1,\;0<c_{j}<s\\ \pi_{2}\left(d_{j},c_{j}\right)=\frac{2}{s^{2}B\left(a,b\right)}\left(s-c_{j}\right)d_{j}^{a-1}{\left(1-d_{j}\right)}^{b-1}, & 0<d_{j}<1,\;0<c_{j}<s\\ \pi_{3}\left(d_{j},c_{j}\right)=\frac{2c_{j}}{s^{2}B\left(a,b\right)}d_{j}^{a-1}{\left(1-d_{j}\right)}^{b-1}, & 0<d_{j}<1,\;0<c_{j}<s \end{array}\right\}.$$

These prior distributions are used to guarantee that $g(\lambda_{j}),j=1,2$ is a decreasing function in $\lambda_{j},j=1,2$. Now, under SE loss function, the E-Bayesian estimates of the parameter $\lambda_{j},j=1,2$ can be obtained from (Equation 11), (Equation 17), and (Equation 19). Using the prior distribution, $\pi_{1}\left(d_{j},c_{j}\right)$ is given by
(20)$$\begin{array}{cc} {\hat{\lambda_{j}}}^{EBS1} & =\int \int_{D}{\hat{\lambda_{j}}}^{BS}\left(d_{j},c_{j}\right)\pi_{1}\left(d_{j},c_{j}\right)dc_{j}dd_{j}\\ \; & =\frac{1}{s}\left(D_{j}+\frac{a}{a+b}\right)\ln\left(1+\frac{s}{T}\right),\;j=1,2. \end{array}$$

Similarly, under SE loss function, the E-Bayesian estimates of $\lambda_{j},j=1,2$ based on $\pi_{2}\left(d_{j},c_{j}\right)$ and $\pi_{3}\left(d_{j},c_{j}\right)$ are given, respectively, by
(21)$${\hat{\lambda_{j}}}^{EBS2}=\frac{2}{s}\left(D_{j}+\frac{a}{a+b}\right)\left[\left(1+\frac{T}{s}\right)\ln\left(1+\frac{s}{T}\right)-1\right],\;j=1,2,$$
and
(22)$${\hat{\lambda_{j}}}^{EBS3}=\frac{2}{s}\left(D_{j}+\frac{a}{a+b}\right)\left[1-\frac{T}{s}\ln\left(1+\frac{s}{T}\right)\right],\;j=1,2.$$

The E-MSE of the parameter $\lambda_{j},j=1,2$ can be obtained from (Equation 15), (Equation 18), and (Equation 19). Using the prior distribution, $\pi_{1}\left(d_{j},c_{j}\right)$ is given by
(23)$$\begin{array}{cc} E-\mathrm{MSE}({\hat{\lambda_{j}}}^{BS1}) & =\int \int_{D}\mathrm{MSE}\left({\hat{\lambda_{j}}}^{BS}\right)\left(d_{j},c_{j}\right)\pi_{1}\left(d_{j},c_{j}\right)dc_{j}dd_{j}\\ & =\left(D_{j}+\frac{a}{a+b}\right)\left(\frac{1}{T(T+s)}\right),\;j=1,2. \end{array}$$

Similarly, under SE loss function, the E-MSE of $\lambda_{j},j=1,2$ based on $\pi_{2}\left(d_{j},c_{j}\right)$ and $\pi_{3}\left(d_{j},c_{j}\right)$ are given, respectively, by
(24)$$E-\mathrm{MSE}\left({\hat{\lambda_{j}}}^{BS2}\right)=\frac{2}{s^{2}}\left(D_{j}+\frac{a}{a+b}\right)\left[\frac{s}{T}-\ln\left(1+\frac{s}{T}\right)\right],\;j=1,2,$$
and
(25)$$E-\mathrm{MSE}\left({\hat{\lambda_{j}}}^{BS3}\right)=\frac{2}{s^{2}}\left(D_{j}+\frac{a}{a+b}\right)\left[\ln\left(1+\frac{s}{T}\right)-\frac{s}{T+s}\right],\;j=1,2.$$

### 5.2. E-Bayesian Estimation under LINEX Loss Function

Under LINEX loss function, the E-Bayesian estimation of $\lambda_{j},j=1,2$ can be obtained by using the different prior distributions of the hyper-parameters given by (Equation 19). For the prior distribution, $\pi_{1}\left(d_{j},c_{j}\right),j=1,2$ and, based on (Equation 14), (Equation 17), and (Equation 19), the E-Bayesian estimate of $\lambda_{j},j=1,2$ is obtained as
(26)$$\begin{array}{cc} {\hat{\lambda_{j}}}^{EBL1} & =\int \int_{D}{\hat{\lambda_{j}}}^{BL}\left(d_{j},c_{j}\right)\pi_{1}\left(d_{j},c_{j}\right)dc_{j}dd_{j}\\ & =\frac{1}{s\omega }\left(D_{j}+\frac{a}{a+b}\right)\left[s\ln\left(1+\frac{\omega }{s+T}\right)+\left(T+\omega \right)\ln\left(1+\frac{s}{T+\omega }\right)-\right.\\ & \;\;\;\;\left.T\ln\left(1+\frac{s}{T}\right)\right],\;j=1,2. \end{array}$$

Similarly, under LINEX loss function, the E-Bayesian estimates of $\lambda_{j},j=1,2$ using $\pi_{2}\left(d_{j},c_{j}\right)$ and $\pi_{3}\left(d_{j},c_{j}\right)$ are given, respectively, by
(27)$$\begin{array}{cc} {\hat{\lambda_{j}}}^{EBL2} & =\left(D_{j}+\frac{a}{a+b}\right)\left[\frac{1}{\omega }\ln\left(1+\frac{\omega }{T}\right)-\frac{{\left(T+s\right)}^{2}}{s^{2}\omega }\ln\left(1+\frac{s}{T}\right)+\frac{{\left(T+\omega +s\right)}^{2}}{s^{2}\omega }\times \right.\\ \; & \left.\ln\left(1+\frac{s}{T+\omega }\right)-\frac{1}{s}\right],\;j=1,2, \end{array}$$
and
(28)$$\begin{array}{cc} {\hat{\lambda_{j}}}^{EBL3} & =\left(D_{j}+\frac{a}{a+b}\right)\left[\frac{1}{\omega }\ln\left(1+\frac{\omega }{s+T}\right)+\frac{T^{2}}{s^{2}\omega }\ln\left(1+\frac{s}{T}\right)-\frac{{\left(T+\omega \right)}^{2}}{s^{2}\omega }\times \right.\\ \; & \left.\ln\left(1+\frac{s}{T+\omega }\right)+\frac{1}{s}\right],\;j=1,2. \end{array}$$

The E-MSE of the parameter $\lambda_{j},j=1,2$ can be obtained from (Equation 16), (Equation 18), and (Equation 19). Using the prior distribution, $\pi_{1}\left(d_{j},c_{j}\right)$ is given by
(29)$$E-\mathrm{MSE}\left({\hat{\lambda_{j}}}^{BLk}\right)=\int \int_{D}\mathrm{MSE}\left({\hat{\lambda_{j}}}^{BL}\right)\left(d_{j},c_{j}\right)\pi_{k}\left(d_{j},c_{j}\right)dc_{j}dd_{j},\;j=1,2,\;k=1,2,3,$$
where ${\hat{\lambda_{j}}}^{BL}$, *j* = 1,2, is the Bayesian estimator of $\lambda_{j}$ given by Equation (Equation 14), $MSE({\hat{\lambda_{j}}}^{BL})$ is the mean square error of Bayesian estimator of $\lambda_{j}$ given by Equation (Equation 16) and *D* is the domain of $d_{j}$ and $c_{j}$ for which the prior density is decreasing in $\lambda_{j}$.

The E-MSE of the parameter $\lambda_{j},j=1,2$ can be obtained from (Equation 14), (Equation 18), and (Equation 19). Using the prior distribution, $\pi_{1}\left(d_{j},c_{j}\right)$ is given by
(30)$$\begin{array}{cc} E-\mathrm{MSE}({\hat{\lambda_{j}}}^{BL1})= & \int \int_{D}\mathrm{MSE}\left({\hat{\lambda_{j}}}^{BL}\right)\left(d_{j},c_{j}\right)\pi_{1}\left(d_{j},c_{j}\right)dc_{j}dd_{j}\\ = & \frac{1}{s}\{\frac{s}{T(T+s)}I_{1j}+\frac{2}{s\omega }I_{2j}\left({\mathrm{Li}}_{2}\left(-\frac{\omega }{T}\right)-{\mathrm{Li}}_{2}\left(-\frac{\omega }{s+T}\right)\right)-\frac{1}{s\omega^{2}}I_{2j}[-\omega \\ & +2\omega \log\left(T\right)\log{\left[T+\omega \right]}^{2}+2\omega \log\left(T+\omega \right)\log\left(\frac{T+\omega }{T}\right)+T\log{\left(\frac{T+\omega }{T}\right)}^{2}\\ & \times \left(\log\left(T+\omega \right)-\log\left(\frac{T+\omega }{\omega }\right)\right)-2\omega \log\left(s+T\right)\log\left(s+T+\omega \right)\\ & +\omega \log{\left(s+T+\omega \right)}^{2}+2\omega \log\left(s+T\right)\log\left(\frac{s+T+\omega }{\omega }\right)-2\omega \log\left(s+T+\omega \right)\\ & \times \log\left(1+\frac{\omega }{s+T}\right)-\left(s+T\right)\log{\left(1+\frac{\omega }{s+T}\right)}^{2}-2\omega {\mathrm{Li}}_{2}\left(-\frac{T}{\omega }\right)\\ & +2\omega {\mathrm{Li}}_{2}\left(-\frac{s+T}{\omega }\right)]\}, \end{array}$$
where ${\mathrm{Li}}_{n}\left(z\right)=\sum_{k=1}^{\infty }\frac{z^{k}}{k^{n}}$, |z|<1 is the polylogarithm function,
$$\begin{array}{c} I_{1j}=D_{j}^{2}+\frac{3a+b}{a+b}D_{j}+\frac{a(2+2a+b)}{\left(a+b)(a+b+1\right)},\;j=1,2, \end{array}$$
and
$$\begin{array}{c} I_{2j}=D_{j}^{2}+\frac{2a}{a+b}D_{j}+\frac{a(a+1)}{\left(a+b)(a+b+1\right)},\;j=1,2. \end{array}$$

Similarly, under LINEX loss function, the E-MSE of $\lambda_{j},j=1,2$ based on $\pi_{2}\left(d_{j},c_{j}\right)$ and $\pi_{3}\left(d_{j},c_{j}\right)$ are given, respectively, by
(31)$$\begin{array}{cc} E-\mathrm{MSE}({\hat{\lambda_{j}}}^{BL2})= & \frac{2I_{1j}}{s^{2}}\left(\frac{s}{T}-\log\left(1+\frac{s}{T}\right)\right)-\frac{4I_{2j}}{s^{2}\omega }\left(\omega \log\left(T+\omega \right)+T\log\left(\frac{T+\omega }{T}\right)\right.\\ & -\omega \log\left(s+T+\omega \right)-\left(s+T\right)\log\left(1+\frac{\omega }{s+T}\right)-\left(s+T\right){\mathrm{Li}}_{2}\left(-\frac{\omega }{T}\right)\\ & +\left(s+T\right){\mathrm{Li}}_{2}\left(-\frac{\omega }{s+T}\right)-\frac{1}{4\omega }\left(\omega \left(2\left(s+T\right)+\omega \right)\log{\left(T+\omega \right)}^{2}+2T\omega \log\left(\frac{T+\omega }{T}\right)\right.\\ & -T\left(2s+T\right)\log{\left(\frac{T+\omega }{T}\right)}^{2}+2\omega \log\left(T+\omega \right)\left(\omega -\left(2\left(s+T\right)+\omega \right)\left(\log(T)\right.\right.\\ & \left.\left.+\log\left(\frac{T+\omega }{T}\right)\right)\right)+\omega \left(2\left(2\left(s+T\right)+\omega \right)\log\left(T\right)\log\left(\frac{T+\omega }{\omega }\right)+\left(-2\omega \right.\right.\\ & \left.+\left(2\left(s+T\right)+\omega \right)\left(2\log(s+T)-\log(s+T+\omega )\right)\right)\log\left(s+T+\omega \right)\\ & \left.-2\left(2\left(s+T\right)+\omega \right)\log\left(s+T\right)\log\left(\frac{s+T+\omega }{\omega }\right)\right)+2\omega \left(-s-T+(2(s+T)+\omega )\right.\\ & \times \left.\log(s+T+\omega )\right)\log\left(1+\frac{\omega }{s+T}\right)+{\left(s+T\right)}^{2}\log{\left(1+\frac{\omega }{s+T}\right)}^{2}\\ & +2\omega \left(2\left(s+T\right)+\omega \right)\left.\left.\left({\mathrm{Li}}_{2}\left(-\frac{T}{\omega }\right)-{\mathrm{Li}}_{2}\left(-\frac{s+T}{\omega }\right)\right)\right)\right). \end{array}$$
(32)$$\begin{array}{cc} E-\mathrm{MSE}({\hat{\lambda_{j}}}^{BL3})= & \frac{2I_{1j}}{s^{2}}\left(\log\left(1+\frac{s}{T}\right)-\frac{s}{s+T}\right)+\frac{4I_{2j}}{s^{2}\omega }\left(\omega \log\left(T+\omega \right)+T\log\left(\frac{T+\omega }{T}\right)\right.\\ & \left.-\omega \log\left(s+T+\omega \right)-\left(s+T\right)\log\left(1+\frac{\omega }{s+T}\right)-T{\mathrm{Li}}_{2}\left(-\frac{\omega }{T}\right)+T{\mathrm{Li}}_{2}\left(-\frac{\omega }{s+T}\right)\right)\\ & +\frac{I_{2j}}{s^{2}\omega^{2}}\left(-\omega \left(2T+\omega \right)\log{\left(T+\omega \right)}^{2}-2T\omega \log\left(\frac{T+\omega }{T}\right)+T^{2}\log{\left(\frac{T+\omega }{T}\right)}^{2}\right.\\ & +2\omega \log\left(T+\omega \right)\left(-\omega +\left(2T+\omega \right)\left(\log\left(T\right)+\log\left(\frac{T+\omega }{T}\right)\right)\right)\\ & +\omega \left(-2\left(2T+\omega \right)\log\left(T\right)\log\left(\frac{T+\omega }{\omega }\right)+\log\left(s+T+\omega \right)\left(2\omega -2(2T+\omega )\log(s+T)\right.\right.\\ & \left.+\left(2T+\omega \right)\log\left(s+T+\omega \right))+2\left(2T+\omega \right)\log\left(s+T\right)\log\left(\frac{s+T+\omega }{\omega }\right)\right)\\ & +2\omega \left(s+T-\left(2T+\omega \right)\log\left(s+T+\omega \right)\right)\log\left(1+\frac{\omega }{s+T}\right)+\left(s-T\right)\left(s+T\right)\\ & \times \left.\log{\left(1+\frac{\omega }{s+T}\right)}^{2}+2\omega \left(2T+\omega \right)\left(-{\mathrm{Li}}_{2}\left(-\frac{T}{\omega }\right)+{\mathrm{Li}}_{2}\left(-\frac{s+T}{\omega }\right)\right)\right). \end{array}$$

## 6. Properties of E-Bayesian Estimation Based on SE Loss Function

Now, the relations among ${\hat{\lambda_{j}}}^{EBSi}$ and E-MSE$\left({\hat{\lambda_{j}}}^{EBSi}\right)$, (j=1,2, i=1,2,3) estimations will be discussed.

**I. Relations among**${\hat{\lambda_{j}}}^{EBSi}$, (j=1,2, i=1,2,3)

**Proposition** **1.**
*Let 0<s<T, a>0, b>0 and ${\hat{\lambda_{j}}}^{EBSi}$, (j=1,2, i=1,2,3) be given by (Equation 20)–(Equation 22). Then, the following inequalities are*
*(i)* 
*${\hat{\lambda_{j}}}^{EBS3}<{\hat{\lambda_{j}}}^{EBS1}<{\hat{\lambda_{j}}}^{EBS2}$.*
*(ii)* 
*$\lim_{T\to \infty }{\hat{\lambda_{j}}}^{EBS1}=\lim_{T\to \infty }{\hat{\lambda_{j}}}^{EBS2}=\lim_{T\to \infty }{\hat{\lambda_{j}}}^{EBS3}$.*



**Proof.** See Appendix A. □

**II. Relations among**${\hat{\lambda_{j}}}^{EBLi}$, (j=1,2, i=1,2,3)

**Proposition** **2.**
*Let 0<s<T, a>0, b>0 and ${\hat{\lambda_{j}}}^{EBLi}$, (j=1,2, i=1,2,3) be given by (Equation 26)–(Equation 28). Then, the following inequalities are*
*(i)* 
*${\hat{\lambda_{j}}}^{EBL3}<{\hat{\lambda_{j}}}^{EBL1}<{\hat{\lambda_{j}}}^{EBL2}$.*
*(ii)* 
*$\lim_{T\to \infty }{\hat{\lambda_{j}}}^{EBL1}=\lim_{T\to \infty }{\hat{\lambda_{j}}}^{EBL2}=\lim_{T\to \infty }{\hat{\lambda_{j}}}^{EBL3}$.*



**Proof.** See Appendix A. □

**III. Relations among E-MSE**$\left({\hat{\lambda_{j}}}^{EBSi}\right)$, (j=1,2, i=1,2,3)

**Proposition** **3.**
*Let 0<s<T, a>0, b>0, and E-MSE$\left({\hat{\lambda_{j}}}^{EBSi}\right)$ (i=1,2,3) be given by (Equation 23)–(Equation 25). Then, the following inequalities are*
*(i)* 
*$E-\mathrm{MSE}\left({\hat{\lambda_{j}}}^{EBS3}\right)<E-\mathrm{MSE}\left({\hat{\lambda_{j}}}^{EBS1}\right)<E-\mathrm{MSE}\left({\hat{\lambda_{j}}}^{EBS2}\right)$.*
*(ii)* 
$\lim_{T\to \infty }E-\mathrm{MSE}\left({\hat{\lambda_{j}}}^{EBS1}\right)=\lim_{T\to \infty }E-\mathrm{MSE}\left({\hat{\lambda_{j}}}^{EBS2}\right)=\lim_{T\to \infty }E-\mathrm{MSE}\left({\hat{\lambda_{j}}}^{EBS3}\right).$



**Proof.** See Appendix A. □

## 7. Numerical Results 

In this section, the numerical discussion is performed on the hypothesis that the scale parameters and common shape parameter are unknown. MATHCAD 2007 was used to perform all calculations. Numerical clarification is made to evaluate these estimates and to clarify the behavior of the proposed methods. We reanalyze the real data set analyzed by Hoel [25], and reused by Kundu et al. [26]. The simulation study used a comparison of performance of the different estimators values and different schemes.

### 7.1. Real Data

In this subsection, we reanalyze a real data set, which was originally reported in Hoel [25] and later by Kundu et al. [26], Pareek et al. [27], and Cramer and Schmiedt [28]. The data were obtained from a laboratory experiment in which male mice received a radiation dose of 300 roentgens at 35 days to 42 days (5–6 weeks) of age. The cause of death for each mouse was determined by reticulum cell sarcoma as cause 1 and other causes of death as cause 2; there were n=77 observations that remain in the analysis.

**Example** **1.**
*Suppose τ=600, the censoring scheme D=25, $R_{1}=R_{2}=\cdots =R_{19}=2$ and $R_{20}=R_{21}=\cdots =R_{D}=0$. The adaptive type I progressive hybrid censored sample from the original data is given as follows: (40,2), (42,2), (51,2), (62,2), (163,2), (179,2), (206,2), (222,2), (228,2), (252,2), (317,1), (324,2), (341,2), (385,2), (399,1), (407,2), (420,2), (517,2), (524,2), (536,1), (554,1), (571,1), (586,1), (586,2), (594,1).*

*The first component of the data denotes the lifetime and the second indicate the reason of failure. There were $D=25,D_{1}=7$ deaths due to reason 1 and $D_{2}=18$ deaths due to reason 2. The maximum likelihood (ML), the Bayesian estimates (BE) for Prior 0: $d_{1}=d_{2}=c_{1}=c_{2}=0$ and Prior 1: $d_{1}=0.5,c_{1}\;=\;1.2,d_{2}\;=\;0.6,c_{2}\;=\;1.3$, and the E-Bayesian estimate (E-BE) are computed for the following initial values β=0.018,ω=1.5,s=15,a=2,b=3, see Table 1.*


**Example** **2.**
*The same data are used, m and $R_{i}$’s are the same as before, while τ=650 instead of τ=600. The adaptive type-I progressive hybrid sample from the original data in this case is given as: (40,2), (42,2), (51,2), (62,2), (163,2), (179,2), (206,2), (222,2), (228,2), (252,2), (317,1), (324,2), (341,2), (385,2), (399,1), (407,2), (420,2), (517,2), (524,2), (536,1), (554,1), (571,1), (586,1), (586,2), (594,1), (619,2), (621,1), (622,2).*

*Here, D=28, $D_{1}=8$ and $D_{2}=20$. The MLE, BEs, and E-BE are computed for the following initial values: $\beta =0.018,d_{1}=0.5,c_{1}=1.2,d_{2}=0.6,c_{2}=1.3,\omega =1.5,s=3,a=2,b=3$, see Table 2.*


From two examples of previous real data, we noticed that:Exact time τ plays an important role in estimating unknown parameters.Based on the SE and LINEX loss functions for Prior 0, we note that MLE performance is very close to BEs and E-BEs.The E-BEs of $\lambda_{1}$ and $\lambda_{2}$ under LINEX loss function with Prior III give smaller E-MSEs.The ordering of performance of estimates of $\lambda_{1}$ and $\lambda_{2}$ in terms of minimum MSEs (E-MSEs) (from best to worst) are E-BEs with prior III, E-BEs with Prior I, E-BEs with prior II, BEs with prior 1, with BEs with prior 0 and MLs.Based on prior 0, it can be seen that the BEs are quite close to the MLEs.Based on LINEX loss function and minimum MSE’s (E-MSE’s), the E-BEs and BEs are better than those under SE loss function; this due to the use of SE which is symmetric loss function and based on the assumption that the loss is the same in any direction.

### 7.2. Simulation Study

In this subsection, we performed a simulation study to evaluate the performance of different methods presented above. The simulation was conducted according to the following steps:For given values, the sample size n=30,50 and different effective number of failures m=5,10 and τ=0.8,1.2.For given values, the parameters $\left(\lambda_{1},\lambda_{2},\beta \right)=\left(0.4,0.6,1.5\right)$ for each case.Determine three different sampling schemes:
Scheme 1: $R_{1}=\cdots =R_{m-1}=0$ and $R_{m}=n-m$,Scheme 2: $R_{1}=\cdots =R_{m-1}=1$ and $R_{m}=n-2m+1$, andScheme 3: $R_{1}=\cdots =R_{m-1}=R_{m}=\left(n-m\right)/m$.Determine $n,m,R_{i}^{\prime }s,T$ and the value of the parameters $\lambda_{j}$, j=1,2.If $x_{m:m:n}<\tau$, the progressive censoring scheme will become $R_{1},R_{2},\cdots ,R_{m},R_{m+1},\cdots ,R_{D}$, where $R_{m}=R_{m+1}=\cdots =R_{D}=0$.All Bayesian estimates are calculated by two types of priests:
Prior 0 described the case of hyper-parameter values, $d_{1}=d_{2}=c_{1}=c_{2}=0$.Prior 1 described the case of hyper-parameter values, $d_{1}=0.8,c_{1}=2,d_{2}=0.6,c_{2}=1$, see Table 3, Table 4, Table 5 and Table 6.Generate the conventional progressive type-I censored sample from the Weibull model according to the method proposed by Kemp and Kemp [29], by using $X={\left(\frac{1}{\lambda_{k}}\log\left(\frac{1}{1-U}\right)\right)}^{1/\beta }$, k=1,2, where *U* is uniform (0,1).The assumption that the number of failures due to each reason of failures is at least one.Under the SE loss function, the estimates ${\hat{\lambda_{j}}}^{BS}$, $MSE({\hat{\lambda_{j}}}^{BS})$, ${\hat{\lambda_{j}}}^{EBSi}$ and $E-\mathrm{MSE}({\hat{\lambda_{j}}}^{BSi})$, j=1,2, and i=1,2,3 are computed from (Equation 11), (Equation 15), (Equation 20)–(Equation 25).Using the LINEX loss function by choosing (ω=1.5) in all the cases, the estimates ${\hat{\lambda_{j}}}^{BL}$, $MSE({\hat{\lambda_{j}}}^{BL})$, ${\hat{\lambda_{j}}}^{EBLi}$ and $E-\mathrm{MSE}({\hat{\lambda_{j}}}^{BLi})$, j=1,2, i=1,2,3 are computed from (Equation 14), (Equation 16), and (Equation 26)–(Equation 32).Repeat the above steps 10,000 times. The average of all 10,000 estimated values from Steps 9–10 are respectively calculated and summarized.The computational results are displayed in Table 3, Table 4, Table 5 and Table 6.

From Table 3, Table 4, Table 5 and Table 6, we have the following observations:The MSE and E-MSE of the different estimators decrease as *n* increases.For fixed *n*, the MSE of the different estimators decreases as *m* increases.For fixed *n* and *m*, the MSE of the different estimators of $\lambda_{1}$ decrease as *T* increases.For fixed *n* and *m*, the MSE of the different estimators of $\lambda_{2}$ increases as *T* increases.As *m* increases for fixed *n*, the MSE and E-MSE decrease.For fixed *n* and *m*, the MSE and E-MSE of $\lambda_{1}$ decrease as *T* increases.For fixed *n* and *m*, the MSE and E-MSE of $\lambda_{2}$ increase as *T* increases.The Bayesian and E-Bayesian estimates of $\lambda_{1}$ and $\lambda_{2}$ perform better than MLEs in terms of minimum MSE.The E-Bayesian estimates of $\lambda_{1}$ and $\lambda_{2}$ have the minimum MSE among all other estimates.The E-Bayesian estimates of $\lambda_{1}$ and $\lambda_{2}$ using prior distribution 3 perform better than other estimates in terms of minimum MSE.The E-Bayesian estimates of $\lambda_{1}$ and $\lambda_{2}$ based on SE loss function under prior distribution 3 have the minimum MSE comparing with all other estimates.The E-posterior risk of E-Bayesian estimation of $\lambda_{1}$ and $\lambda_{2}$ using prior distribution 3 under LINEX loss function have the minimum values among all other prior distributions.

Combining all the above results, we recommend using the E-Bayesian procedure to estimate the parameters $\lambda_{1}$ and $\lambda_{2}$ for the Adaptive Type-I Progressive Hybrid Censoring scheme based on the prior distribution 3 which performs better than other estimates in terms of minimum MSE and E-MSE.

## 8. Conclusions

In this article, we have investigated the E-Bayesian estimation of the parameter and the reliability function of the Weibull distribution based on A-I PHCS. The E-Bayesian estimation is considered by using three different prior distributions under two loss functions, namely the SE and LINEX loss functions. The properties of the E-Bayesian estimation as well as the E-posterior risk are also derived. We compare the performance of the E-Bayesian estimation with the maximum likelihood and Bayesian estimators via an extensive simulation study. The simulation results reveal that the E-Bayesian estimation performs better than the maximum likelihood and Bayesian estimators in terms of minimum biases and MSEs. Moreover, we analyze two real data sets for illustration purposes, and the results coincide with those in the simulation part.

## Figures and Tables

**Table 1 entropy-22-00903-t001:** The MLE, BE, E-BE, and the MSE(E-MSE).

			BE		E-BE		
${\hat{\lambda }}_{j}$	MLE	LF	Prior 0	Prior 1	Prior I	Prior II	Prior III
${\hat{\lambda }}_{1}$		SE	0.08217456	0.08682112	0.08001569	0.08217776	0.07785363
(MSE)	0.08217456		(9.8767× 10${}^{-4}$)	(9.8669× 10${}^{-4}$)	(8.6710× 10${}^{-4}$)	(9.1394× 10${}^{-4}$)	(8.2026× 10${}^{-4}$)
	(2.0258× 10${}^{-3}$)	LINEX	0.08145945	0.08607594	0.07937234	0.08149984	0.07724485
			(9.8718× 10${}^{-4}$)	(9.8623× 10${}^{-4}$)	(8.6652× 10${}^{-4}$)	(9.1340× 10${}^{-4}$)	(8.1064× 10${}^{-4}$)
${\hat{\lambda }}_{2}$		SE	0.21130602	0.21415876	0.19895794	0.20433388	0.193582
(MSE)	0.21130602		(2.5806× 10${}^{-3}$)	(2.4868× 10${}^{-3}$)	(2.1560× 10${}^{-3}$)	(2.2725× 10${}^{-3}$)	(2.0396× 10${}^{-3}$)
	(3.2831× 10${}^{-3}$)	LINEX	0.20946715	0.21232066	0.19735826	0.20264825	0.19206827
			(2.5739× 10${}^{-3}$)	(2.4802× 10${}^{-3}$)	(2.1486× 10${}^{-3}$)	(2.2654× 10${}^{-3}$)	(2.0319× 10${}^{-3}$)

**Table 2 entropy-22-00903-t002:** The MLE, BE, E-BE, and the MSE(E-MSE).

			BE		E-BS		
${\hat{\lambda }}_{j}$	MLE	LF	Prior 0	Prior 1	Prior I	Prior II	Prior III
${\hat{\lambda }}_{1}$		SE	0.09389191	0.09837467	0.090809	0.093262	0.088356
(MSE)	0.09389191		(1.1219× 10${}^{-3}$)	(1.1177× 10${}^{-3}$)	(9.8386× 10${}^{-4}$)	(1.0369× 10${}^{-3}$)	(9.3072× 10${}^{-4}$)
	(2.0889× 10${}^{-3}$)	LINEX	0.09307501	0.09753052	0.090079	0.092493	0.087665
			(1.1216× 10${}^{-3}$)	(1.1174× 10${}^{-3}$)	(9.8339× 10${}^{-4}$)	(1.0366× 10${}^{-3}$)	(9.3020× 10${}^{-4}$)
${\hat{\lambda }}_{2}$		SE	0.23472978	0.23725655	0.220536	0.226494	0.214579
(MSE)	0.23472978		(2.7549× 10${}^{-3}$)	(2.7529× 10${}^{-3}$)	(2.3894× 10${}^{-3}$)	(2.5184× 10${}^{-3}$)	(2.2603× 10${}^{-3}$)
	(3.5343× 10${}^{-3}$)	LINEX	0.23268753	0.23522066	0.218764	0.224626	0.212901
			(2.7491× 10${}^{-3}$)	(2.7471× 10${}^{-3}$)	(2.3825× 10${}^{-3}$)	(2.5119× 10${}^{-3}$)	(2.2532× 10${}^{-3}$)

**Table 3 entropy-22-00903-t003:** The average values of the MLE, BE, E-BE, and the MSE(E-MSE) in parentheses, under different censoring schemes and different τ’s, for n=30 and m=5.

				BE		E-BE		
*S*	${\hat{\lambda }}_{j}$	MLE	LF	Prior 0	Prior 1	Prior I	Prior II	Prior III
τ=0.8								
			SE	0.07113	0.09168	0.06427	0.0702	0.05923
	${\hat{\lambda }}_{1}$	0.07113		(2.3712× 10${}^{-3}$)	(2.3652× 10${}^{-3}$)	(1.6893× 10${}^{-3}$)	(1.9745× 10${}^{-3}$	(1.4041× 10${}^{-3}$)
	(MSE)	(2.7518× 10${}^{-3}$)	LINEX	0.06941	0.0896	0.06348	0.06876	0.0582
1				(2.3546× 10${}^{-3}$)	(2.3498× 10${}^{-3}$)	(1.67115× 10${}^{-3}$)	(1.9569× 10${}^{-3}$)	(1.3854× 10${}^{-3}$)
			SE	0.09552	0.1118	0.08341	0.09048	0.07634
	${\hat{\lambda }}_{2}$	0.09552		(3.1837× 10${}^{-3}$)	(3.6059× 10${}^{-3}$)	(2.1769× 10${}^{-3}$)	(2.5444× 10${}^{-3}$)	(1.8095× 10${}^{-3}$)
	(MSE)	(NA)	LINEX	0.09321	0.10918	0.08182	0.08863	0.07502
				(3.1694× 10${}^{-3}$)	(3.5931× 10${}^{-3}$)	(2.1599× 10${}^{-3}$)	(2.5283× 10${}^{-3}$)	(1.7915× 10${}^{-3}$)
			SE	0.0714	0.09197	0.06492	0.07043	0.05941
	${\hat{\lambda }}_{1}$	0.0714		(2.3846× 10${}^{-3}$)	(2.3792× 10${}^{-3}$)	(1.69710× 10${}^{-3}$)	(1.9840× 10${}^{-3}$)	(1.4102× 10${}^{-3}$)
	(MSE)	(4.1442× 10${}^{-3}$)	LINEX	0.06967	0.08988	0.06368	0.06898	0.05838
2				(2.3681× 10${}^{-3}$)	(2.3639× 10${}^{-3}$)	(1.6789× 10${}^{-3}$)	(1.9665× 10${}^{-3}$)	(1.3915× 10${}^{-3}$)
			SE	0.09556	0.11186	0.08342	0.09051	0.07634
	${\hat{\lambda }}_{2}$	0.09556		(3.1908× 10${}^{-3}$)	(3.1144× 10${}^{-3}$)	(2.1806× 10${}^{-3}$)	(2.549× 10${}^{-3}$)	(1.8119× 10${}^{-3}$)
	(MSE)	(NA)	LINEX	0.09324	0.10923	0.08183	0.08865	0.07502
				(3.1766× 10${}^{-3}$)	(3.1017× 10${}^{-3}$)	(2.1635× 10${}^{-3}$)	(2.5331× 10${}^{-3}$)	(1.7939× 10${}^{-3}$)
			SE	0.07173	0.09243	0.06514	0.07070	0.05958
	${\hat{\lambda }}_{1}$	0.07173		(2.4134× 10${}^{-3}$)	(2.3135× 10${}^{-3}$)	(1.7133× 10${}^{-3}$)	(2.0045× 10${}^{-3}$)	(1.4219× 10${}^{-3}$)
	(MSE)	(4.1634× 10${}^{-3}$)	LINEX	0.06998	0.09031	0.06389	0.06924	0.05854
3				(2.3969× 10${}^{-3}$)	(2.3084× 10${}^{-3}$)	(1.6952× 10${}^{-3}$)	(1.9871× 10${}^{-3}$)	(1.4033× 10${}^{-3}$)
			SE	0.09648	0.11287	0.08407	0.09124	0.07689
	${\hat{\lambda }}_{2}$	0.09648		(3.2457× 10${}^{-3}$)	(3.1735× 10${}^{-3}$)	(2.2109× 10${}^{-3}$)	(2.5867× 10${}^{-3}$)	(1.8349× 10${}^{-3}$)
	(MSE)	(NA)	LINEX	0.09412	0.11020	0.08245	0.08936	0.07555
				(3.2317× 10${}^{-3}$)	(3.1609× 10${}^{-3}$)	(2.1939× 10${}^{-3}$)	(2.5708× 10${}^{-3}$)	(1.8170× 10${}^{-3}$)
τ=1.2								
			SE	0.07108	0.09164	0.06468	0.07017	0.0592
	${\hat{\lambda }}_{1}$	0.07108		(2.3696× 10${}^{-3}$)	(2.3638× 10${}^{-3}$)	(1.6884× 10${}^{-3}$)	(1.9734× 10${}^{-3}$)	(1.4033× 10${}^{-3}$)
	(MSE)	(NA)	LINEX	0.06936	0.08956	0.06345	0.06873	0.05817
1				(2.3531× 10${}^{-3}$)	(2.3485× 10${}^{-3}$)	(1.6702× 10${}^{-3}$)	(1.9559× 10${}^{-3}$)	(1.3846× 10${}^{-3}$)
			SE	0.09557	0.11184	0.08345	0.09052	0.07638
	${\hat{\lambda }}_{2}$	0.09557		(3.1853× 10${}^{-3}$)	(3.1674× 10${}^{-3}$)	(2.1779× 10${}^{-3}$)	(2.5455× 10${}^{-3}$)	(1.8103× 10${}^{-3}$)
	(MSE)	(6.7849× 10${}^{-2}$)	LINEX	0.09326	0.10922	0.08186	0.08867	0.07505
				(3.1711× 10${}^{-3}$)	(3.1547× 10${}^{-3}$)	(2.1609× 10${}^{-3}$)	(2.5295× 10${}^{-3}$)	(1.7923× 10${}^{-3}$)
			SE	0.07117	0.09176	0.06474	0.07024	0.05925
	${\hat{\lambda }}_{1}$	0.07117		(2.3770× 10${}^{-3}$)	(2.3725× 10${}^{-3}$)	(1.6925× 10${}^{-3}$)	(1.9787× 10${}^{-3}$)	(1.4064× 10${}^{-3}$)
	(MSE)	(NA)	LINEX	0.06945	0.08967	0.06351	0.0688	0.05822
2				(2.3604× 10${}^{-3}$)	(2.3572× 10${}^{-3}$)	(1.6744× 10${}^{-3}$)	(1.9611× 10${}^{-3}$)	(1.3877× 10${}^{-3}$)
			SE	0.09579	0.11208	0.0836	0.0907	0.0765
	${\hat{\lambda }}_{2}$	0.09579		(3.1986× 10${}^{-3}$)	(3.1627× 10${}^{-3}$)	(2.1852× 10${}^{-3}$)	(2.5546× 10${}^{-3}$)	(1.8158× 10${}^{-3}$)
	(MSE)	(5.0019× 10${}^{-3}$)	LINEX	0.09347	0.10945	0.0820	0.08883	0.07517
				(3.1844× 10${}^{-3}$)	(3.1609× 10${}^{-3}$)	(2.1682× 10${}^{-3}$)	(2.5386× 10${}^{-3}$)	(1.7979× 10${}^{-3}$)
			SE	0.07194	0.09262	0.0653	0.07088	0.05973
	${\hat{\lambda }}_{1}$	0.07194		(2.4206× 10${}^{-3}$)	(2.4199× 10${}^{-3}$)	(1.7176× 10${}^{-3}$)	(2.0096× 10${}^{-3}$)	(1.4255× 10${}^{-3}$)
	(MSE)	(4.1995× 10${}^{-3}$)	LINEX	0.07018	0.0905	0.06405	0.06941	0.05868
3				(2.4042× 10${}^{-3}$)	(2.4041× 10${}^{-3}$)	(1.6995× 10${}^{-3}$)	(1.9922× 10${}^{-3}$)	(1.4068× 10${}^{-3}$)
			SE	0.09628	0.11268	0.08392	0.09108	0.07675
	${\hat{\lambda }}_{2}$	0.09628		(3.2392× 10${}^{-3}$)	(3.1674× 10${}^{-3}$)	(2.2069× 10${}^{-3}$)	(2.5822× 10${}^{-3}$)	(1.8317× 10${}^{-3}$)
	(MSE)	(NA)	LINEX	0.09393	0.11001	0.0823	0.0892	0.07541
				(3.2252× 10${}^{-3}$)	(3.1549× 10${}^{-3}$)	(2.2189× 10${}^{-3}$)	(2.5662× 10${}^{-3}$)	(1.8138× 10${}^{-3}$)

**Table 4 entropy-22-00903-t004:** The average values of the MLE, BE, E-BE, and the MSE(E-MSE) in parentheses, under different censoring schemes and different τ’s, for n=30 and m=10.

				BE		E-BE		
*S*	${\hat{\lambda }}_{j}$	MLE	LF	Prior 0	Prior 1	Prior I	Prior II	Prior III
τ=0.8								
			SE	0.14286	0.15895	0.11966	0.12982	0.10951
	${\hat{\lambda }}_{1}$	0.14286		(4.7681× 10${}^{-3}$)	(4.6730× 10${}^{-3}$)	(3.1266× 10${}^{-3}$)	(3.6549× 10${}^{-3}$)	(2.5983× 10${}^{-3}$)
	(MSE)	(7.3873× 10${}^{-3}$)	LINEX	0.1394	0.15534	0.11738	0.12715	0.10761
1				(4.7609× 10${}^{-3}$)	(4.6667× 10${}^{-3}$)	(3.1126× 10${}^{-3}$)	(3.6429× 10${}^{-3}$)	(2.5824× 10${}^{-3}$)
			SE	0.19086	0.20407	0.15643	0.16971	0.14316
	${\hat{\lambda }}_{2}$	0.19086		(6.3693× 10${}^{-3}$)	(6.2941× 10${}^{-3}$)	(4.0869× 10${}^{-3}$)	(4.7775× 10${}^{-3}$)	(3.3965× 10${}^{-3}$)
	(MSE)	(7.6945× 10${}^{-3}$)	LINEX	0.18624	0.19928	0.15345	0.16622	0.14067
				(6.3715× 10${}^{-3}$)	(6.2940× 10${}^{-3}$)	(4.0771× 10${}^{-3}$)	(4.7708× 10${}^{-3}$)	(3.3833× 10${}^{-3}$)
			SE	0.14365	0.15975	0.1202	0.13043	0.10997
	${\hat{\lambda }}_{1}$	0.14365		(4.8142× 10${}^{-3}$)	(4.0175× 10${}^{-3}$)	(3.1512× 10${}^{-3}$)	(3.6854× 10${}^{-3}$)	(2.6171× 10${}^{-3}$)
	(MSE)	(5.1985× 10${}^{-3}$)	LINEX	0.14016	0.1561	0.1179	0.12775	0.10805
2				(4.8073× 10${}^{-3}$)	(4.0115× 10${}^{-3}$)	(3.1374× 10${}^{-3}$)	(3.6735× 10${}^{-3}$)	(2.6012× 10${}^{-3}$)
			SE	0.19146	0.20471	0.1568	0.17014	0.14345
	${\hat{\lambda }}_{2}$	0.19146		(6.4159× 10${}^{-3}$)	(6.3375× 10${}^{-3}$)	(4.1103× 10${}^{-3}$)	(4.8070× 10${}^{-3}$)	(3.4136× 10${}^{-3}$)
	(MSE)	(2.1338× 10${}^{-3}$)	LINEX	0.1868	0.19989	0.15379	0.16664	0.14095
				(6.4185× 10${}^{-3}$)	(6.3215× 10${}^{-3}$)	(4.1005× 10${}^{-3}$)	(4.8005× 10${}^{-3}$)	(3.4005× 10${}^{-3}$)
			SE	0.14398	0.16012	0.12038	0.13066	0.1101
	${\hat{\lambda }}_{1}$	0.14398		(4.8463× 10${}^{-3}$)	(4.0498× 10${}^{-3}$)	(3.1672× 10${}^{-3}$)	(3.7059× 10${}^{-3}$)	(2.6286× 10${}^{-3}$)
	(MSE)	(5.6554× 10${}^{-3}$)	LINEX	0.14046	0.15645	0.11807	0.12796	0.10817
3				(4.8395× 10${}^{-3}$)	(4.0439× 10${}^{-3}$)	(3.1535× 10${}^{-3}$)	(3.6942× 10${}^{-3}$)	(2.6128× 10${}^{-3}$)
			SE	0.19261	0.20587	0.15757	0.17102	0.14411
	${\hat{\lambda }}_{2}$	0.19261		(6.4829× 10${}^{-3}$)	(6.4238× 10${}^{-3}$)	(4.1455× 10${}^{-3}$)	(4.8505× 10${}^{-3}$)	(3.4404× 10${}^{-3}$)
	(MSE)	(6.6906× 10${}^{-3}$)	LINEX	0.1879	0.2010	0.15454	0.16749	0.14159
				(6.4859× 10${}^{-3}$)	(6.4183× 10${}^{-3}$)	(4.1358× 10${}^{-3}$)	(4.8442× 10${}^{-3}$)	(3.4274× 10${}^{-3}$)
τ=1.2								
			SE	0.14344	0.15949	0.12011	0.1303	0.10992
	${\hat{\lambda }}_{1}$	0.14344		(4.7873× 10${}^{-3}$)	(4.6898× 10${}^{-3}$)	(3.1381× 10${}^{-3}$)	(3.6683× 10${}^{-3}$)	(2.6079× 10${}^{-3}$)
	(MSE)	(3.3292× 10${}^{-3}$)	LINEX	0.13997	0.15587	0.11782	0.12762	0.10801
1				(4.7802× 10${}^{-3}$)	(4.6837× 10${}^{-3}$)	(3.1242× 10${}^{-3}$)	(3.6564× 10${}^{-3}$)	(2.5919× 10${}^{-3}$)
			SE	0.19028	0.20351	0.15599	0.16922	0.14275
	${\hat{\lambda }}_{2}$	0.19028		(6.3498× 10${}^{-3}$)	(6.2721× 10${}^{-3}$)	(4.0753× 10${}^{-3}$)	(4.7639× 10${}^{-3}$)	(3.3868× 10${}^{-3}$)
	(MSE)	(7.3309× 10${}^{-3}$)	LINEX	0.18567	0.19873	0.15301	0.16575	0.14027
				(6.3519× 10${}^{-3}$)	(6.2656× 10${}^{-3}$)	(4.0653× 10${}^{-3}$)	(4.7571× 10${}^{-3}$)	(3.3735× 10${}^{-3}$)
			SE	0.14403	0.16011	0.12049	0.13075	0.11023
	${\hat{\lambda }}_{1}$	0.14403		(4.8272× 10${}^{-3}$)	(4.0289× 10${}^{-3}$)	(3.1589× 10${}^{-3}$)	(3.6945× 10${}^{-3}$)	(2.6235× 10${}^{-3}$)
	(MSE)	(5.0914× 10${}^{-3}$)	LINEX	0.14053	0.15645	0.11818	0.12805	0.10831
2				(4.8203× 10${}^{-3}$)	(4.0229× 10${}^{-3}$)	(3.1452× 10${}^{-3}$)	(3.6827× 10${}^{-3}$)	(2.6077× 10${}^{-3}$)
			SE	0.1911	0.20436	0.15652	0.16984	0.14319
	${\hat{\lambda }}_{2}$	0.1911		(6.4044× 10${}^{-3}$)	(6.5267× 10${}^{-3}$)	(4.1033× 10${}^{-3}$)	(4.7989× 10${}^{-3}$)	(3.4078× 10${}^{-3}$)
	(MSE)	(6.3308× 10${}^{-3}$)	LINEX	0.18645	0.19955	0.15352	0.16634	0.1407
				(6.4069× 10${}^{-3}$)	(6.5306× 10${}^{-3}$)	(4.0935× 10${}^{-3}$)	(4.7923× 10${}^{-3}$)	(3.3946× 10${}^{-3}$)
			SE	0.14393	0.16008	0.12034	0.13062	0.11006
	${\hat{\lambda }}_{1}$	0.14393		(4.8448× 10${}^{-3}$)	(4.0485× 10${}^{-3}$)	(3.1663× 10${}^{-3}$)	(3.7048× 10${}^{-3}$)	(2.6278× 10${}^{-3}$)
	(MSE)	(3.5418× 10${}^{-3}$)	LINEX	0.14041	0.15641	0.11803	0.12792	0.10814
3				(4.8380× 10${}^{-3}$)	(4.0427× 10${}^{-3}$)	(3.1525× 10${}^{-3}$)	(3.6931× 10${}^{-3}$)	(2.6119× 10${}^{-3}$)
			SE	0.19267	0.20593	0.15761	0.17107	0.14415
	${\hat{\lambda }}_{2}$	0.19267		(6.4850× 10${}^{-3}$)	(6.4747× 10${}^{-3}$)	(4.1467× 10${}^{-3}$)	(4.8519× 10${}^{-3}$)	(3.4414× 10${}^{-3}$)
	(MSE)	(6.8266× 10${}^{-3}$)	LINEX	0.18796	0.20106	0.15458	0.16753	0.14163
				(6.4880× 10${}^{-3}$)	(6.4713× 10${}^{-3}$)	(4.1370× 10${}^{-3}$)	(4.8457× 10${}^{-3}$)	(3.4284× 10${}^{-3}$)

**Table 5 entropy-22-00903-t005:** The average values of the MLE, BE, E-BE, and the MSE(E-MSE) in parentheses, under different censoring scheme and different τ’s, for n=50 and m=5.

				BE		E-BE		
*S*	${\hat{\lambda }}_{j}$	MLE	LF	Prior 0	Prior 1	Prior I	Prior II	Prior III
τ=0.8								
			SE	0.04273	0.05646	0.04267	0.04506	0.04029
	${\hat{\lambda }}_{1}$	0.04273		(8.5401× 10${}^{-4}$)	(8.3850× 10${}^{-4}$)	(7.2425× 10${}^{-4}$)	(8.0513× 10${}^{-4}$)	(6.4337× 10${}^{-4}$)
	(MSE)	(1.4851× 10${}^{-3}$)	LINEX	0.0421	0.05566	0.04214	0.04447	0.03981
1				(8.5247× 10${}^{-4}$)	(8.6570× 10${}^{-4}$)	(7.2259× 10${}^{-4}$)	(8.0354× 10${}^{-4}$)	(6.4164× 10${}^{-4}$)
			SE	0.05719	0.06782	0.05483	0.0579	0.05177
	${\hat{\lambda }}_{2}$	0.05719		(1.4270× 10${}^{-3}$)	(1.3287× 10${}^{-3}$)	(7.3054× 10${}^{-4}$)	(8.0344× 10${}^{-4}$)	(7.2664× 10${}^{-4}$)
	(MSE)	(NA)	LINEX	0.05635	0.06684	0.05415	0.05714	0.05116
				(1.1235× 10${}^{-3}$)	(1.0968× 10${}^{-3}$)	(7.29066× 10${}^{-4}$)	(8.0151× 10${}^{-4}$)	(7.2505× 10${}^{-4}$)
			SE	0.04275	0.0565	0.04269	0.0450	0.0403
	${\hat{\lambda }}_{1}$	0.04275		(8.5542× 10${}^{-4}$)	(8.456× 10${}^{-4}$)	(7.2528× 10${}^{-4}$)	(8.0635× 10${}^{-4}$)	(6.4422× 10${}^{-4}$)
	(MSE)	(NA)	LINEX	0.04213	0.0557	0.04216	0.04449	0.03983
2				(8.5388× 10${}^{-4}$)	(8.4756× 10${}^{-4}$)	(7.2362× 10${}^{-4}$)	(8.0476× 10${}^{-4}$)	(6.4248× 10${}^{-4}$)
			SE	0.05727	0.06791	0.0549	0.05798	0.05183
	${\hat{\lambda }}_{2}$	0.05727		(1.4565× 10${}^{-3}$)	(1.3320× 10${}^{-3}$)	(9.3260× 10${}^{-4}$)	(1.0368× 10${}^{-3}$)	(8.2838× 10${}^{-4}$)
	(MSE)	(2.4449× 10${}^{-3}$)	LINEX	0.05643	0.06693	0.05422	0.05721	0.05122
				(2.1264× 10${}^{-3}$)	(1.3129× 10${}^{-3}$)	(9.3113× 10${}^{-4}$)	(1.0175× 10${}^{-3}$)	(8.2679× 10${}^{-4}$)
			SE	0.04316	0.05701	0.04304	0.04547	0.04061
	${\hat{\lambda }}_{1}$	0.04316		(1.7145× 10${}^{-3}$)	(1.1063× 10${}^{-3}$)	(7.3691× 10${}^{-4}$)	(8.1991× 10${}^{-4}$)	(6.5391× 10${}^{-4}$)
	(MSE)	(NA)	LINEX	0.04252	0.0562	0.0425	0.04486	0.04013
3				(1.7172× 10${}^{-3}$)	(1.0871× 10${}^{-3}$)	(7.3526× 10${}^{-4}$)	(8.1833× 10${}^{-4}$)	(6.5218× 10${}^{-4}$)
			SE	0.05777	0.0685	0.05531	0.05843	0.05219
	${\hat{\lambda }}_{2}$	0.05777		(1.6633× 10${}^{-3}$)	(1.3556× 10${}^{-3}$)	(9.4698× 10${}^{-4}$)	(1.0536× 10${}^{-3}$)	(8.4033× 10${}^{-4}$)
	(MSE)	(2.4824× 10${}^{-3}$)	LINEX	0.05692	0.06751	0.05461	0.05766	0.05157
				(1.1471× 10${}^{-3}$)	(1.3366× 10${}^{-3}$)	(9.4553× 10${}^{-4}$)	(1.0343× 10${}^{-3}$)	(8.3876× 10${}^{-4}$)
τ=1.2								
			SE	0.04253	0.05627	0.04251	0.04489	0.04013
	${\hat{\lambda }}_{1}$	0.04253		(8.5009× 10${}^{-3}$)	(1.0814× 10${}^{-3}$)	(7.2145× 10${}^{-4}$)	(8.0201× 10${}^{-4}$)	(6.4088× 10${}^{-4}$)
	(MSE)	(9.0935× 10${}^{-3}$)	LINEX	0.04191	0.05548	0.04198	0.04429	0.03966
1				(8.4853× 10${}^{-3}$)	(1.0621× 10${}^{-3}$)	(7.1978× 10${}^{-4}$)	(8.0041× 10${}^{-4}$)	(6.3914× 10${}^{-4}$)
			SE	0.05738	0.06801	0.0550	0.05808	0.05192
	${\hat{\lambda }}_{2}$	0.05738		(1.1466× 10${}^{-3}$)	(1.1325× 10${}^{-3}$)	(9.3335× 10${}^{-4}$)	(1.0376× 10${}^{-3}$)	(8.2913× 10${}^{-4}$)
	(MSE)	(NA)	LINEX	0.05654	0.06703	0.05431	0.05731	0.05131
				(1.1274× 10${}^{-3}$)	(1.1135× 10${}^{-3}$)	(9.3188× 10${}^{-4}$)	(1.0182× 10${}^{-3}$)	(8.2755× 10${}^{-4}$)
			SE	0.04278	0.05652	0.04271	0.0451	0.04032
	${\hat{\lambda }}_{1}$	0.04278		(8.5592× 10${}^{-3}$)	(1.0873× 10${}^{-3}$)	(7.2564× 10${}^{-4}$)	(8.0674× 10${}^{-4}$)	(6.4453× 10${}^{-4}$)
	(MSE)	(1.4880× 10${}^{-3}$)	LINEX	0.04215	0.05572	0.04218	0.04451	0.03985
2				(8.5437× 10${}^{-3}$)	(1.0680× 10${}^{-3}$)	(7.2397× 10${}^{-4}$)	(8.0515× 10${}^{-4}$)	(6.4279× 10${}^{-4}$)
			SE	0.05725	0.06789	0.05489	0.05796	0.05181
	${\hat{\lambda }}_{2}$	0.05725		(1.4526× 10${}^{-3}$)	(1.3316× 10${}^{-3}$)	(9.3232× 10${}^{-4}$)	(1.0365× 10${}^{-3}$)	(8.2813× 10${}^{-4}$)
	(MSE)	(NA)	LINEX	0.05641	0.06692	0.0542	0.0572	0.0512
				(1.3603× 10${}^{-3}$)	(1.3126× 10${}^{-3}$)	(9.3085× 10${}^{-4}$)	(1.0172× 10${}^{-3}$)	(8.2654× 10${}^{-4}$)
			SE	0.04314	0.05699	0.04302	0.04545	0.04059
	${\hat{\lambda }}_{1}$	0.04314		(8.7087× 10${}^{-3}$)	(1.1058× 10${}^{-3}$)	(7.3649× 10${}^{-4}$)	(8.1944× 10${}^{-4}$)	(6.5355× 10${}^{-4}$)
	(MSE)	(NA)	LINEX	0.0425	0.05617	0.04248	0.04484	0.04011
3				(8.6934× 10${}^{-3}$)	(1.0965× 10${}^{-3}$)	(7.3484× 10${}^{-4}$)	(8.1786× 10${}^{-4}$)	(6.5182× 10${}^{-4}$)
			SE	0.05779	0.06852	0.05533	0.05845	0.0522
	${\hat{\lambda }}_{2}$	0.05779		(1.1665× 10${}^{-3}$)	(1.1557× 10${}^{-3}$)	(9.4711× 10${}^{-4}$)	(1.0538× 10${}^{-3}$)	(8.4046× 10${}^{-4}$)
	(MSE)	(2.4932× 10${}^{-3}$)	LINEX	0.05693	0.06752	0.05463	0.05767	0.05158
				(1.1473× 10${}^{-3}$)	(1.1368× 10${}^{-3}$)	(9.4566× 10${}^{-4}$)	(1.0344× 10${}^{-3}$)	(8.3888× 10${}^{-4}$)

**Table 6 entropy-22-00903-t006:** The average values of the MLE, BE, E-BE, and the MSE(E-MSE) in parentheses, under different censoring scheme and different τ’s, for n=50 and m=10.

				BE		E-BE		
*S*	${\hat{\lambda }}_{j}$	MLE	LF	Prior 0	Prior 1	Prior I	Prior II	Prior III
τ=0.8								
			SE	0.08591	0.09799	0.0790	0.08342	0.07458
	${\hat{\lambda }}_{1}$	0.08591		(1.7174× 10${}^{-3}$)	(1.6835× 10${}^{-3}$)	(1.3411× 10${}^{-3}$)	(1.4909× 10${}^{-3}$)	(1.1913× 10${}^{-3}$)
	(MSE)	(1.7436× 10${}^{-3}$)	LINEX	0.08465	0.0966	0.07801	0.08232	0.0737
1				(1.6991× 10${}^{-3}$)	(1.6655× 10${}^{-3}$)	(1.3222× 10${}^{-3}$)	(1.4722× 10${}^{-3}$)	(1.1721× 10${}^{-3}$)
			SE	0.11398	0.12351	0.10262	0.10836	0.09687
	${\hat{\lambda }}_{2}$	0.11398		(2.27844× 10${}^{-3}$)	(2.2205× 10${}^{-3}$)	(1.7419× 10${}^{-3}$)	(1.9364× 10${}^{-3}$)	(1.5473× 10${}^{-3}$)
	(MSE)	(4.7343× 10${}^{-3}$)	LINEX	0.11231	0.12173	0.10133	0.10693	0.09573
				(2.2614× 10${}^{-3}$)	(2.2037× 10${}^{-3}$)	(1.7237× 10${}^{-3}$)	(1.9186× 10${}^{-3}$)	(1.5287× 10${}^{-3}$)
			SE	0.08613	0.09822	0.07917	0.08361	0.07473
	${\hat{\lambda }}_{1}$	0.08613		(1.7261× 10${}^{-3}$)	(1.6926× 10${}^{-3}$)	(1.3469× 10${}^{-3}$)	(1.4976× 10${}^{-3}$)	(1.1691× 10${}^{-3}$)
	(MSE)	(1.7476× 10${}^{-3}$)	LINEX	0.08486	0.09683	0.07817	0.08250	0.07385
2				(1.7078× 10${}^{-3}$)	(1.6747× 10${}^{-3}$)	(1.3279× 10${}^{-3}$)	(1.4789× 10${}^{-3}$)	(1.1769× 10${}^{-3}$)
			SE	0.11427	0.12382	0.10284	0.1086	0.09707
	${\hat{\lambda }}_{2}$	0.11427		(2.290× 10${}^{-3}$)	(2.2325× 10${}^{-3}$)	(1.7495× 10${}^{-3}$)	(1.9453× 10${}^{-3}$)	(1.5537× 10${}^{-3}$)
	(MSE)	(2.7584× 10${}^{-3}$)	LINEX	0.11259	0.12203	0.10155	0.10717	0.09592
				(2.2729× 10${}^{-3}$)	(2.2158× 10${}^{-3}$)	(1.7313× 10${}^{-3}$)	(1.9275× 10${}^{-3}$)	(1.5351× 10${}^{-3}$)
			SE	0.08657	0.09874	0.0795	0.08399	0.07501
	${\hat{\lambda }}_{1}$	0.08657		(1.7484× 10${}^{-3}$)	(1.7167× 10${}^{-3}$)	(1.3615× 10${}^{-3}$)	(1.5149× 10${}^{-3}$)	(1.2082× 10${}^{-3}$)
	(MSE)	(2.2551× 10${}^{-3}$)	LINEX	0.08529	0.09733	0.0785	0.08287	0.07412
3				(1.7301× 10${}^{-3}$)	(1.6988× 10${}^{-3}$)	(1.3427× 10${}^{-3}$)	(1.4963× 10${}^{-3}$)	(1.1890× 10${}^{-3}$)
			SE	0.11538	0.12497	0.10369	0.10955	0.09784
	${\hat{\lambda }}_{2}$	0.11538		(2.3300× 10${}^{-3}$)	(2.2738× 10${}^{-3}$)	(1.7759× 10${}^{-3}$)	(1.9759× 10${}^{-3}$)	(1.5758× 10${}^{-3}$)
	(MSE)	(2.7765× 10${}^{-3}$)	LINEX	0.11366	0.12315	0.10239	0.10809	0.09668
				(2.3130× 10${}^{-3}$)	(2.2572× 10${}^{-3}$)	(1.7577× 10${}^{-3}$)	(1.9582× 10${}^{-3}$)	(1.5573× 10${}^{-3}$)
τ=1.2								
			SE	0.08561	0.09769	0.07874	0.08315	0.07434
	${\hat{\lambda }}_{1}$	0.08561		(1.7113× 10${}^{-3}$)	(1.6778× 10${}^{-3}$)	(1.3367× 10${}^{-3}$)	(1.4859× 10${}^{-3}$)	(1.1874× 10${}^{-3}$)
	(MSE)	(NA)	LINEX	0.08435	0.09631	0.07776	0.08205	0.07346
1				(1.6929× 10${}^{-3}$)	(1.6598× 10${}^{-3}$)	(1.3178× 10${}^{-3}$)	(1.4673× 10${}^{-3}$)	(1.1682× 10${}^{-3}$)
			SE	0.11429	0.1238	0.10287	0.10862	0.09711
	${\hat{\lambda }}_{2}$	0.11429		(2.2844× 10${}^{-3}$)	(2.2262× 10${}^{-3}$)	(1.7462× 10${}^{-3}$)	(1.9412× 10${}^{-3}$)	(1.5512× 10${}^{-3}$)
	(MSE)	(2.7105× 10${}^{-3}$)	LINEX	0.11261	0.12202	0.10158	0.10719	0.09597
				(2.2673× 10${}^{-3}$)	(2.2095× 10${}^{-3}$)	(1.7279× 10${}^{-3}$)	(1.9235× 10${}^{-3}$)	(1.5325× 10${}^{-3}$)
			SE	0.08613	0.09822	0.07917	0.08361	0.07473
	${\hat{\lambda }}_{1}$	0.08613		(1.7261× 10${}^{-3}$)	(1.6926× 10${}^{-3}$)	(1.3469× 10${}^{-3}$)	(1.4976× 10${}^{-3}$)	(1.1961× 10${}^{-3}$)
	(MSE)	(2.4756× 10${}^{-3}$)	LINEX	0.08486	0.09683	0.07817	0.0825	0.07385
2				(1.7078× 10${}^{-3}$)	(1.6747× 10${}^{-3}$)	(1.3279× 10${}^{-3}$)	(1.4789× 10${}^{-3}$)	(1.1769× 10${}^{-3}$)
			SE	0.11427	0.12382	0.10284	0.1086	0.09707
	${\hat{\lambda }}_{2}$	0.11427		(2.2900× 10${}^{-3}$)	(2.2325× 10${}^{-3}$)	(1.7495× 10${}^{-3}$)	(1.9453× 10${}^{-3}$)	(1.5537× 10${}^{-3}$)
	(MSE)	(2.7584× 10${}^{-3}$)	LINEX	0.11259	0.12203	0.1015	0.10717	0.09592
				(2.2729× 10${}^{-3}$)	(2.2158× 10${}^{-3}$)	(1.7313× 10${}^{-3}$)	(1.9275× 10${}^{-3}$)	(1.5351× 10${}^{-3}$)
			SE	0.08642	0.09859	0.07937	0.08385	0.07489
	${\hat{\lambda }}_{1}$	0.08642		(1.7454× 10${}^{-3}$)	(1.7139× 10${}^{-3}$)	(1.3594× 10${}^{-3}$)	(1.5125× 10${}^{-3}$)	(1.2062× 10${}^{-3}$)
	(MSE)	(1.9812× 10${}^{-3}$)	LINEX	0.08513	0.09718	0.07837	0.08274	0.0740
3				(1.7271× 10${}^{-3}$)	(1.659× 10${}^{-3}$)	(1.3405× 10${}^{-3}$)	(1.4939× 10${}^{-3}$)	(1.1871× 10${}^{-3}$)
			SE	0.11554	0.12513	0.10383	0.10969	0.09797
	${\hat{\lambda }}_{2}$	0.11534		(2.3334× 10${}^{-3}$)	(2.2771× 10${}^{-3}$)	(1.7783× 10${}^{-3}$)	(1.9786× 10${}^{-3}$)	(1.5779× 10${}^{-3}$)
	(MSE)	(2.6643× 10${}^{-3}$)	LINEX	0.11382	0.12331	0.10252	0.10823	0.09681
				(2.31647× 10${}^{-3}$)	(2.2605× 10${}^{-3}$)	(1.7601× 10${}^{-3}$)	(1.9609× 10${}^{-3}$)	(1.5594× 10${}^{-3}$)

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
