# Peer review of "E-Bayesian Estimation for the Weibull Distribution under Adaptive Type-I Progressive Hybrid Censored Competing Risks Data"

_entropy, 2020, doi:10.3390/e22080903_

Round 1

Reviewer 1 Report

Please read the report attached. English writing needs to be polished, specially, the real data set illustration part.

No methodological errors; but some notations possibly need to check.   

Reviewer 2 Report

The authors present e-bayesian estimation under an adaptive censoring scheme. A weibull model and a linex loss function are used. The paper is well written and results appear correct.

In my opinion the paper lacks some motivation as it appears the authors simply develop some already existing theory for cases not yet considered in the literature. There is no really new ideas from the methods or methodological point of view.

I think the authors could try discussing in the introduction and the real data examples why the choices they made are important in relation to what is already available in the literature.

For example, linex loss is asymmetric. What are the applications where this could be relevant? In the examples they provide,  which advantages, in terms of understanding of the phenomenon, will the linex loss  bring?

Author Response

Entropy-865460

E-Bayesian Estimation for the Weibull Distribution under Adaptive Type-I Progressive Hybrid Censored Competing Risks Data,

We thank the referees for many constructive comments and suggestions that have greatly improved the paper. Please find below our responses (in regular font) to each of the comments and suggestions (in bold). We feel that the revised version is a significant improvement of the previous one.

Reviewer #2

The authors present E-Bayesian estimation under an adaptive censoring scheme. A Weibull model and a linex loss function are used. The paper is well written and results appear correct.

In my opinion the paper lacks some motivation as it appears the authors simply develop some already existing theory for cases not yet considered in the literature. There is no really new ideas from the methods or methodological point of view.

I think the authors could try discussing in the introduction and the real data examples why the choices they made are important in relation to what is already available in the literature.

For example, linex loss is asymmetric. What are the applications where this could be relevant? In the examples they provide, which advantages, in terms of understanding of the phenomenon, will the linex loss bring?

Authors:

 Response:  It was clarified by adding a paragraph in the introduction. Authors

Round 2

Reviewer 1 Report

Authors have revised almost all of the items listed for the orginal draft submitted, except the following items:

  1. The Definition of non-informative prior from Line 113 to Line 115 is right.  The prior must be a uniform distribution i.e. g(\theta) = constant.  But your example, g(\theta) = 1/\theta is not a uniform distribution because this funtion is descreasing function.  Please remove "non-informative prior" from all statements because you do not use uniform prior.
  2. Line 102, is \gamma_{i}=m-i+1-\sum_{j=1}^{m} R_{j} right? 
  3. Line 104, is J=D1 + D_{2} right?

Author Response

Reviewer 1

Comment # 1:  The Definition of non-informative prior from Line 113 to Line 115 is right.  The prior must be a uniform distribution i.e. g (\theta) = constant.  But your example, g (\theta) = 1/\theta is not a uniform distribution because this function is decreasing function.  Please remove "non-informative prior" from all statements because you do not use uniform prior.

Answer:  Thank you very much. This paragraph was deleted and we have deleted the use of “non-informative” in the new version of the manuscript.

Comment # 2:  Line 102, is \gamma_{i}=m-i+1-\sum_{j=1}^{m} R_ {j} right? 

Answer:  Many thanks, it is changed to “c is a constant doesn’t depend on the parameters”.

Comment # 3:  Line 104, is J=D1 + D_ {2} right? 

Answer:  Many thanks, It is corrected to J=D_1+D_2.

Reviewer 2 Report

Lines 74-89. Linex loss.This part is quite confused with repetitions of the same concepts, typos (e.g. "detailed explanation" and not "detail eplanation") and vague technical explanation. Better just give  a concise introduction with references and postpone more technical discussion about parameters after introducing formally the loss function.

Lines 113-117. Remark 1. I think we could give for granted the concept of non-informative prior. The explanation provided is quite awkward:note that the prior g(\lambda) = 1/lambda is NOT uniform as the density is decreasing. To understand better the problem they are discussing the authors should compare non-informative priors for scale and location parameters.

Line 222. Real examples. I suppose the object is to estimate the survival function. Can this be made more explicit? Why in this problem Underestimation (overestimation) of parameters is more important than (overestimation) (underestimation)? I.e. what does Linex loss add to the analysis, apart getting a smaller MSE in some cases?

Author Response

Reviewer 2

 Comment # 1:  Lines 74-89. Linex loss.This part is quite confused with repetitions of the same concepts, typos (e.g. "detailed explanation" and not "detail eplanation") and vague technical explanation. Better just give a concise introduction with references and postpone more technical discussion about parameters after introducing formally the loss function. ---

Answer:  Thank you very much. This paragraph was changed.

Comment # 2:  Lines 113-117. Remark 1. I think we could give for granted the concept of non-informative prior. The explanation provided is quite awkward: note that the prior g(\lambda) = 1/lambda is NOT uniform as the density is decreasing. To understand better the problem they are discussing the authors should compare non-informative priors for scale and location parameters. -

Answer:  Thank you very much. This paragraph was deleted and we have deleted the use of “non-informative” in the new version of the manuscript.

Comment # 3:  Line 222. Real examples. I suppose the object is to estimate the survival function. Can this be made more explicit? Why in this problem Underestimation (overestimation) of parameters is more important than (overestimation) (underestimation)? I.e. what does Linex loss add to the analysis, apart getting a smaller MSE in some cases?

Answer:  

  • They explained how the LINEX loss function worked, but there were no details or practical explanations given on how the LINEX loss function works in changing the shape parameter and the error function. For more detail “ Khatun, M. A. Matin, A Study on LINEX Loss Function with Different Estimating Methods, 10 (2020) 52-63”.

  • In Bayesian estimation, some methods are always depending on loss data, and this method is compared to the method of calculating the so-called average square error, and the lower its value, this indicates that this method is better than others in this case.

This manuscript is a resubmission of an earlier submission. The following is a list of the peer review reports and author responses from that submission.